# Angiogenesis and Tissue Repair Depend on Platelet Dosing and Bioformulation Strategies Following Orthobiological Platelet-Rich Plasma Procedures: A Narrative Review

**DOI:** 10.3390/biomedicines11071922

**Published:** 2023-07-06

**Authors:** Peter A. Everts, José Fábio Lana, Kentaro Onishi, Don Buford, Jeffrey Peng, Ansar Mahmood, Lucas F. Fonseca, Andre van Zundert, Luga Podesta

**Affiliations:** 1Research & Education Division, Gulf Coast Biologics, Fort Myers, FL 33916, USA; 2OrthoRegen Group, Max-Planck University, Indaiatuba, São Paulo 13334-170, Brazil; josefabiolana@gmail.com; 3Department of Orthopaedics, The Bone and Cartilage Institute, Indaiatuba, São Paulo 13334-170, Brazil; 4Department of PM&R and Orthopedic Surgery, University of Pittsburg Medical Center, Pittsburgh, PA 15213, USA; kenonishi918@gmail.com; 5Texas Orthobiologics, Dallas, TX 75204, USA; donbufordmd@gmail.com; 6Stanford Health Care—O’Connor Hospital Sports Medicine, Stanford University School of Medicine, San Jose, CA 95128, USA; jpeng@stanfordhealthcare.org; 7Department of Trauma and Orthopaedic Surgery, University Hospitals, Birmingham B15 2GW, UK; ansar@doctors.org.uk; 8Department of Orthopaedics, The Federal University of São Paulo, São Paulo 04024-002, Brazil; ffonsecalu@gmail.com; 9Department of Anaesthesia and Perioperative Medicine, Royal Brisbane and Women’s Hospital, Brisbane and the University of Queensland, Brisbane 4072, Australia; vanzundertandre@gmail.com; 10Bluetail Medical Group & Podesta Orthopedic Sports Medicine, Naples, FL 34109, USA; lugamd@aol.com

**Keywords:** angiogenesis, tissue repair, platelet-rich plasma, platelet dose, bioformulation, leukocytes, platelet-rich fibrin, orthobiology, biosurgery

## Abstract

Angiogenesis is the formation of new blood vessel from existing vessels and is a critical first step in tissue repair following chronic disturbances in healing and degenerative tissues. Chronic pathoanatomic tissues are characterized by a high number of inflammatory cells; an overexpression of inflammatory mediators; such as tumor necrosis factor-α (TNF-α) and interleukin-1 (IL-1); the presence of mast cells, T cells, reactive oxygen species, and matrix metalloproteinases; and a decreased angiogenic capacity. Multiple studies have demonstrated that autologous orthobiological cellular preparations (e.g., platelet-rich plasma (PRP)) improve tissue repair and regenerate tissues. There are many PRP devices on the market. Unfortunately, they differ greatly in platelet numbers, cellular composition, and bioformulation. PRP is a platelet concentrate consisting of a high concentration of platelets, with or without certain leukocytes, platelet-derived growth factors (PGFs), cytokines, molecules, and signaling cells. Several PRP products have immunomodulatory capacities that can influence resident cells in a diseased microenvironment, inducing tissue repair or regeneration. Generally, PRP is a blood-derived product, regardless of its platelet number and bioformulation, and the literature indicates both positive and negative patient treatment outcomes. Strangely, the literature does not designate specific PRP preparation qualifications that can potentially contribute to tissue repair. Moreover, the literature scarcely addresses the impact of platelets and leukocytes in PRP on (neo)angiogenesis, other than a general one-size-fits-all statement that “PRP has angiogenic capabilities”. Here, we review the cellular composition of all PRP constituents, including leukocytes, and describe the importance of platelet dosing and bioformulation strategies in orthobiological applications to initiate angiogenic pathways that re-establish microvasculature networks, facilitating the supply of oxygen and nutrients to impaired tissues.

## 1. Introduction

Over the past two decades, autologous biological cellular products prepared at the point of care have become a growing area of innovative medical healthcare treatments, such as non-surgical interventional biological treatments. These autologous cellular materials include platelet-rich plasma (PRP) preparations, bone marrow concentrate, and various adipose tissue preparations [1]. These preparations include various cells, such as platelets, mesenchymal stem cells (MSCs), leukocytes, and signaling cells, and typical platelet growth factors (PGFs), cytokines, proteinases, chemokines, and interleukins (ILs) [1].

In addition to conventional therapies, non-surgical interventional biological methods can treat conditions where conventional therapies are ineffective or insufficient. A major concern following orthobiological procedures in chronic pathological tissues is the inability to restore an adequate microvascular blood vessel network for oxygen diffusion and nutrient uptake. Therefore, orthobiological PRP preparations that promote vascularization by inducing angiogenesis (i.e., the growth of new capillaries from preexisting vessels) may play an important role in tissue repair and regeneration. However, the role of various PRP constituents in angiogenic pathways and blood vessel formation has been poorly described. Thus far, no standards have been established for biological preparations [2,3,4]. As a result, many devices have been introduced to the market, most of which lack supporting clinical data on their biological effectiveness and clinical outcomes, particularly regarding the cellular functions in PRP and their effects on tissue repair and (neo)angiogenesis.

Because of a lack of consensus, some clinical studies have had adverse outcomes or failed to yield positive results. Based on the literature, there appears to be a correlation between administered platelet doses and therapeutic outcomes, supporting the concept that classical and angiogenesis cascades ultimately contribute to the regeneration of functional tissues [5,6,7]. Interestingly, the angiogenic cascade and the commonalities with the classical healing cascade in tissue repair and their relationship to platelet dosing and bioformulation strategies in PRP preparations [8] are infrequently discussed, even though the establishment of a vascular network is crucial to the regeneration of damaged tissues and allows nutrients, oxygen, and waste products to exchange [9]. A clear and fundamental understanding of the molecular mechanisms that regulate angiogenesis should allow the development of PRP platelet dosing and bioformulation strategies in orthobiological procedures.

This narrative review characterizes the properties of platelets and leukocyte cellular contents in PRP composition in neo-angiogenesis. We examine the effects of platelet dosing, multicellular PRP biomolecules, and platelet angiogenic factors that can modulate angiogenic processes and blood vessel formation, which can ultimately lead to tissue repair. A better understanding of the physiology and factors that affect angiogenesis will lead to more consistent and successful outcomes for patients receiving PRP.

## 2. Angiogenesis in Tissue Repair

Tissue regeneration and wound healing involve restorative events, including cell homing, resident and circulating cells, and the release of soluble mediators generated by the extracellular matrix [10,11]. The optimal healing process involves a cascade of complex, overlapping, and well-orchestrated four-phasic events, with intra-platelet granules playing an early role; restoring the damaged blood supply to damaged tissues is an essential component of tissue repair [12]. At the onset of the inflammatory phase, the tissue healing response requires angiogenesis, a process by which endothelial cells (ECs) are stimulated, proliferate, and migrate to form new blood vessels and tight junctions from preexisting vessels [13].

### 2.1. Overview of Angiogenic Processes

Vasculogenesis is the de novo formation of new blood vessels from precursor cells (e.g., angioblasts) that differentiate into ECs, which ultimately become primitive blood vessels. In contrast, angiogenesis, or neovascularization, is the formation of new blood vessels from an existing vascular bed and is one of the key processes in tissue growth and repair. This process involves a physiological sequence consisting of vasodilatation, the degradation of basement cell migration, chemotaxis, increasing vascular permeability, EC proliferation, vessel formation, and a reduced risk of cell death. Consequently, angiogenesis is a regulatory mechanism for tissue regeneration and repair, delivering oxygen, micronutrients, and inflammatory cells to the injured area [8]. Unlike angiogenesis, arteriogenesis, or collateralization, refers to the formation of new blood vessels caused by hypertrophy and luminal distention of preexisting vessels as a result of redirected blood flow from occluded or stenosed distal vessels [14].

### 2.2. Functional Mechanism of Angiogenesis

Angiogenesis is a critical element of many regenerative and reparative processes, such as acute wound healing, as well as pathologic processes. In both situations, new blood vessel development is driven by the physiological condition in a local microenvironment and is tightly regulated by the balance between various angiogenic regulatory molecules (e.g., endogenous stimulators and inhibitors) that regulate the physiological control of angiogenesis [15].

Stimulatory and inhibitory molecules are stored in platelets and inflammatory cells. Platelets are sturdy angiogenic influencers as they release both pro-angiogenic factors and anti-angiogenetic factors (Table 1). ECs are actively regulated by angiogenic growth factors that stimulate proliferation, migration, tube formation, and maturation at locations where angiogenesis is required, whereas negative regulators are a diverse group of molecules that prevent ECs from undergoing these processes. Damaged tissues that require greater vascularity and perfusion produce more angiogenic growth factors, tipping the balance of regulation in favor of blood vessel growth. A precise physiological balance between pro-angiogenic and anti-angiogenic factors is critical in homeostasis, tissue development, and tissue repair [16].

Interestingly, platelets, monocytes, and macrophages are associated with tissue repair, releasing angiogenic growth factors into injured regions. In severely injured tissues, apoptotic cells release intracellular angiogenic stimulatory factors, with the development of newly formed capillaries (i.e., arteriogenesis) [17]. Steady-state angiogenic suppression is achieved once the conditions for vascular growth are met by reducing growth factor production, resolving inflammation, and increasing local concentrations of angiogenesis inhibitors [18].

### 2.3. Angiogenesis in Acute and Chronic Tissue Repair

To achieve consistent positive patient outcomes following non-surgical orthobiological regenerative medicine techniques, it is paramount that the physiological process by which new blood vessels form via angiogenesis is quickly re-established in degenerative or traumatized tissue structures, as blood vessels allow for oxygen and micronutrient delivery to the tissues while removing catabolic waste products [19]. The rationale for employing PRP and other autologous biological products is that they can mimic the classical healing and immune system cascades, starting with the initial platelet clot. This process involves the growth of new capillaries to form granulation and de novo tissue. This new scaffold acts as a matrix for cell proliferation, cell migration, and fibroblast–collagen synthesis, followed by the other healing phases for full tissue restoration and functional repair [12,20,21].

However, in many chronic musculoskeletal (MSK) disorders comprising diverse pathologic conditions affecting joints, tendons, ligaments, and other connective tissue, tissue repair and regeneration following the classical healing cascade phases is not a valid hypothesis. Tissue repair through this cascade is delayed and frequently results in chronic inflammation due to the prolonged presence of neutrophils, macrophages of phenotype 1 (M1), and mast cell degranulation [22]. The numbers of T helper (Th) cells (e.g., Th1, Th17, and Th22) and matrix metalloproteinases (MMPs) are also increased under chronic pathological conditions [23]. All these pathological processes promote chronic inflammation, tissue fibrosis, and poor tissue vascularity and impede angiogenesis [22]. The foremost function of PRP therapies in chronic MSK maladies should be restoring the vascular architecture, which will facilitate the high metabolic activity of tissue repair and regeneration processes. This approach is of substantial clinical importance in pathoanatomic disorders in areas of poor vascularization, such as meniscal tears, labrum pathologies, and tendon or ligament injuries [24,25].

In chronic pathologies, the reinstitution of blood flow enables the recruitment of reparative MSCs from the bone marrow via peripheral circulation. Increased angiogenic potential facilitates the migration of bone-marrow-derived cells to the PRP-treated tissue sites, where bone marrow cells can proliferate, differentiate, and carry out progenitor cell processes to ultimately repair tissue [26].

It is interesting to note that in many chronic and degenerative tendinopathies, new blood vessels develop in the area of the pathology [27,28]. It is paradoxical that despite neoangiogenesis being associated with tissue healing in diseased tissues, this hypervascularity following chronic tendinosis is not associated with tissue healing [29]. Indeed, Tempfer and Traweger have suggested that increased aberrant vascularity is responsible for impaired tendon tissue repair in chronic tendon pathologies [30].

### 2.4. Biosurgery: Implementing PRP in Surgical Techniques

Although beyond the scope of this review, it is worth mentioning that PRP, in particular activated platelet-rich plasma gel (PRP-G), has successfully been utilized in orthopedic, cardio-vascular, maxillofacial, and plastic-reconstructive surgeries to optimize post-surgical tissue healing [31]. Biosurgical procedures embrace different technologies and surgical techniques intended to support post-surgical tissue wound healing by facilitating hemostasis, and tissue sealing by using autologously prepared biologics, like PRP and MSC preparations [32,33]. Mixing PRP with calcium chloride and thrombin formulations, will transform liquid PRP into a scaffold-like, malleable, transient structure, referred to as PRP-G, shown in Figure 1 [34].

The polymerization of plasma proteins present in a PRP fraction produces a three-dimensional cross-linked fibrin matrix, binding non-activated platelets and leukocytes. PRP-G has been successfully used in many MSK-related surgical procedures, resulting in fewer post-surgical complications, with improved functional outcome scores, and faster return to play in professional athletes [36,37,38,39].

The rationale to employ bio-surgical procedures is that after the local and homogenous delivery of PRP-G to surgically repaired tissue sites, the PRP-G cloth will integrate and subsequently interact directly with the ECM of the repaired tissue structures. In this acute surgical tissue repair model, naturally occurring fibrinolysis will gradually degradate the PRP-G scaffold, facilitating a steady, continuous, sustained, and localized release of PGFs and other cytokine concentrations, for up to 28 days [40]. Consequently, released PRP-G platelet constituents, molecules, proteins, and leukocytic cells will induce cell proliferation, differentiation, and migration [41]. Furthermore, PRP-G constituents can regulate angiogenetic pathways and stimulate the natural healing cascade to govern repair mechanisms to reconstruct tissue structures and ultimately restore function [42]. Improved biosurgical outcomes have been attributed to the angiogenetic potential of PRP-G by adequately restoring the blood supply in repaired tissues and in surgically dissected tissues and supporting surgical wound healing. Consequently, angiogenesis lowered the duration and/or intensity of post-surgical pain [19,43]. Importantly, since diabetic, obese, and plastic-reconstructive skin grafting patients are at risk for poor post-operative outcomes and wound healing disturbances, the development of a functional angiogenetic cascade in the acute tissue repair process is imperative.

## 3. Role and Interactions of the Extracellular Matrix in Angiogenesis

It has long been believed that the extracellular matrix (ECM) holds tissues together by acting as a glue; however, numerous studies have demonstrated that it also plays a significant role in cellular behavior and complex biological processes through a wide variety of cell–cell mechanisms, including the morphogenesis of vascular vessels [44]. This latter process requires several interactions between vascular cells, signaling molecules, and the extracellular environment to facilitate angiogenesis. The ECM controls the availability of platelet growth factors and modulates the signaling by growth factors through their interactions with receptors, ECM components, and cell-adhesion receptors [45]. Furthermore, growth factors induce greater or less cell signaling, depending on the ECM surrounding the growth factor receptors.

The ECM plays a central role in tissue repair and wound healing; however, it also plays a role in morphogenesis, embryogenesis, and (neo)angiogenesis [46]. Importantly, cell–ECM interactions impact and regulate vascular morphogenic processes, including sprouting and non-sprouting angiogenesis, blood vessel assembly, maturation, and vascular remodeling [47]. Specifically, the ECM modulates and binds to various PGFs to regulate vascular morphogenesis [48]. Various vascular endothelial growth factor (VEGF) isoforms bind to matrix molecules after splicing by plasminogen and MMPs, resulting in clear ECM protein affinities [49]. As a result of these endoproteolytic events, soluble and matrix-bound VEGF levels in tissues are modulated, and matrix-bound VEGF is primarily responsible for inducing sprouting angiogenesis [50]. The ECM has a functional role in angiogenesis, as specific growth factors are sequestered in the ECM as reservoirs that cells release on demand, resembling a sustained release phenomenon.

## 4. Analytical Background PRP Technology

Autologous PRP refers to a concentrated liquid fraction of freshly derived, anticoagulated peripheral blood whose platelet count exceeds baseline values, currently defined as exceeding 1 × 10^6^/µL [3]. PRP specimens can be prepared with simple test-tube-like kits using a low blood volume or with a more refined process using larger blood volumes. The latter process often uses dual-spin centrifugation systems that can produce multicellular products, including leukocytic fractions [2]. PRP is generally characterized by its absolute platelet concentration after minimal manipulative non-traumatic centrifugation. Following the two-step centrifugation procedure, whole blood cellular density separation results in the concentration of platelets and potentially other cells based on the cellular densities of the individual blood cells. For all types of PRP products, platelets should be the primary cells present in the final product [2,4].

### 4.1. PRP in Tissue Repair Mechanisms

PRP therapies are based on the hypothesis that injections of supraphysiologic platelet numbers at degenerative or injured tissue sites result in the release of many biologically active factors, mimicking natural healing processes [51]. PRP therapies offer the advantage of being an autologous therapy with no systemic adverse effects, other than occasionally reported pain and swelling following treatment, compared to non-autologous biologics [52,53]. Because of these safety characteristics, autologous PRP has been incorporated into a wide range of clinical indications, demonstrating clinical benefits and promising patient outcomes [54,55,56].

Analytical proteomic studies performed on some PRP products have demonstrated that the cellular activities performed by the various platelet constituents within the storage granules play an important role in maintaining most basic cell functions by acting as endocrine, paracrine, autocrine, and intracrine mediators [57,58]. Several studies have indicated the potential of platelets and PRP in angiogenesis by enhancing EC proliferation, migration, and tube formation [59,60].

### 4.2. PRP Device Variables and Considerations

Many orthobiological PRP injectates are available on the market, which vary significantly in quality and efficacy. These device variations and preparation protocols account for the varying cell yields in the PRP specimens [61]. The platelet concentrations obtained by some systems are even lower than the baseline platelet concentrations in the whole blood, whereas dual-spin closed systems can produce PRP specimens containing platelets exceeding 1.6 × 10^6^/µL [2]. Few PRP systems on the market can produce high platelet concentrations and allow the preparation of diverse bioformulations based on leukocyte and erythrocyte counts [62]. To provide a high dose of PRP platelets, the patient’s baseline platelet concentration, pre-donated blood volume, and PRP device capture rate must be considered and addressed accordingly.

Oh et al. recognized great variations in cellular composition, concentrations, and biomolecular characteristics among different PRP preparations. Their controlled laboratory study found that dual-spin methods were more likely to produce high PRP platelet concentrations than single-spin methods. In addition, VEGF and PDGF were significantly higher following dual-spin PRP preparation than following single-spin methods because dual-spin devices have dramatically higher platelet capture rates than single-spin devices.

### 4.3. Transitioning from Platelet Concentration to Platelet Dosing

In the scientific literature, the most frequently mentioned quality parameter for PRP applications is the increase in PRP platelet numbers above the whole blood baseline values. However, the fold increase over baseline values is only a quality marker for the efficacy of the PRP device used and the executed preparation procedure. Additionally, the platelet fold increase over baseline is an invalid parameter to correlate patient outcomes following a PRP procedure. Concisely, the concentration of platelets or multiples of platelets above baseline do not describe the total number of platelets that have be delivered to pathoanatomic tissue sites. Therefore, multiplying the actual platelet concentration by the injected PRP volume reflects the total delivered platelets to a treatment site and is synonymous with the term “platelet dose” [25]. The platelet dose should be measured in billions or millions of platelets. The DEPA classification proposed the following dosing definitions: very high dose (>5 billion platelets), high dose (3–5 billion platelets), medium dose (1–3 billion platelets), and low dose (<1 billion platelets) [63]. This insinuates that the actual therapeutic platelet dose can be immensely different, and therefore, so can the availability of angiogenetic platelet factors. Furthermore, it has been demonstrated that cell repair processes respond to PRP in a platelet-dose-dependent manner [5].

Therefore, we propose to consider adopting the platelet dose as a new and better PRP quality parameter, as this delineates how many platelets must be delivered to a single treatment site to induce angiogenesis and tissue repair. Transitioning from the PRP platelet concentration to platelet dosing as a quality parameter has the clear advantage that patient outcomes can be correlated with an effective therapeutic platelet dose, as mentioned for the first time by Bansal et al. Importantly, clinicians can effectively reproduce PRP treatment protocols in order to achieve similar tissue repair outcomes, including consistent analgesic effects [64,65].

### 4.4. Gravitational Cellular Density Separation and Platelet Dose

Dual-spin PRP devices enable more optimized PRP preparations with higher platelet concentrations because they exploit the principles of gravitational cellular density separation, resulting in a layered buffy-coat stratum, as visualized in Figure 2 [66]. As a result of centrifugation, blood cells are concentrated and separated based on their size and density differences. The concentrated cells are arranged in an orderly manner in a small volume of plasma at the bottom of the PRP device, from which platelets and other cellular constituents can be withdrawn as a PRP concentrate. Comparatively, single-spin test tubes create a product from the acellular plasma layer, which excludes all leukocytes and erythrocytes.

The PRP device platelet capture rate, available PRP volume, and associated whole blood preparation volume are critical factors in manufacturing PRP and they vary greatly between devices, leading to lower platelet concentrations and, therefore, the potential to implement platelet dosing strategies in orthobiological treatments is inconceivable [4]. Several studies have shown that the PRP platelet dose is important in determining patient outcomes [67,68]. Moreover, several in vitro studies have shown that PRP stimulates cell–cell interactions in a dose-dependent manner [5,69]. Considering that all the available platelets and other PRP constituents interact with resident and recipient cells to facilitate tissue repair and angiogenic processes, low platelet dosages will likely adversely impact angiogenic processes, and therefore tissue repair.

**Figure 2 biomedicines-11-01922-f002:**
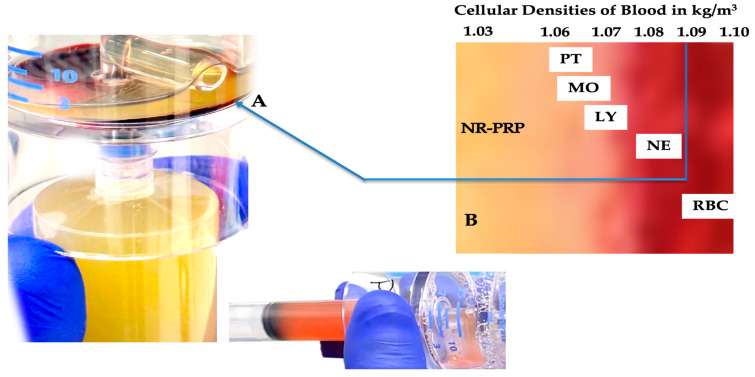
Whole blood cellular gravitational density separation after a dual-spin procedure. In this picture, a 60 mL PRP device (PurePRP-SP^®^, EmCyte Corporation, Fort Myers FL, USA, with permission) was used to produce LR-PRP. At A, a thin, gray, multicomponent, buffy-coat layer is visible. Before extraction of this layer, platelets are resuspended in a small volume of plasma and aspirated in a syringe. Resuspending the platelets colors the PRP slightly light amber, indicating a capture of the buffy-coat layer. A magnified image of the buffy-coat layer in B illustrates how blood cells are arranged according to their densities [70]. To realize an LR-PRP specimen, a fraction of RBCs must be part of the bioformulation, as the density of neutrophils overlaps with the density of the RBCs. In this 3.75 mL LR-PRP product, the total available platelets were 10.02 × 10^9^; monocyte and neutrophil concentrations were 7.95 and 9.92 × 10^3^/µL, respectively. The RBC concentration was 2.01 × 10^6^/µL.

## 5. Clinical PRP: A Myriad of Angiogenic Stimulators and Actors

Platelets are well known for their critical role in primary hemostasis. Increasing evidence has demonstrated that these cells are important modulators of other physio pathological processes, including angiogenesis, inflammation, and immunomodulation [3]. Angiogenic stimulators originate from many different sources, including platelets, ECs, fibroblasts, smooth muscle cells, leukocytes, and inflammatory cells [71,72]. The importance and potential for platelets to release angiogenic regulatory proteins, platelet-derived growth factors (PDGFs), cytokines, and chemokines are well established [73,74], as shown in several well-executed in vitro and animal studies.

### 5.1. Platelet Dosing and Bioformulation Are Important Factors in Angiogenesis Regulation

The pivotal role of platelets in angiogenesis has been widely acknowledged; however, strategizing PRP dosing and bioformulations for chronic pathoanatomic tissues has scarcely been explored in orthobiology [25], except as a general concept. It has been repeatedly proclaimed that PRP has angiogenic capabilities, which positions it as a “one-size-fits-all” autologous biological product.

Neglecting the ability to prepare tissue-specific orthobiological PRP products with customized platelet doses and bioformulations limits the potential for angiogenic tissue responses and tissue repair. This failure may explain why variable and conflicting data have been published regarding PRP therapy. Strikingly, single-spin PRP tubes with low volume and low platelet capture rates have been used most frequently in these studies. Thus, the PRP platelet concentration and the presence of leukocytes significantly impact the restoration of the tissue microvasculature and tissue repair mechanisms, which can positively impact patient outcomes.

### 5.2. Platelets

Platelets are small, anucleate discoid blood cells (1–3 mm) generated by megakaryocytes in the red bone marrow and released continuously into the peripheral circulation in a resting state [75]. Platelet counts in adults range from 150 to 350 × 10^6^/mL. Platelets have extracellular glycoprotein receptors, adhesion molecules, and three intra-platelet structures: α-granules, dense granules, and lysosomes, each with specific cellular and molecular contents (Figure 3) [25]. Platelets stimulate proliferation, the synthesis of mesenchymal and neurotrophic factors, chemotactic cell migration, immunomodulatory activities, and angiogenesis [15,76]. Giusti et al. determined that a dose of platelets should contain 1.5 × 10^6^ platelets/µL to stimulate tissue repair mechanisms and induce an angiogenic response via ECs, whereas higher platelet concentrations reduce the platelet’s angiogenic capacity in follicular and perifollicular angiogenesis [77].

After PRP delivery to degenerated or traumatized tissues (e.g., ligaments and tendons), platelets are activated through tissue factor mechanisms to release their granular content [78], such as granular PGF and cytokines [1,11].

### 5.3. Platelet α-Granules

Platelets have organelles containing three types of secretory granules, including α-granules that store PGF and other angiogenic regulatory proteins—a mixture of proteins synthesized by megakaryocytes or endocytosed from circulation by megakaryocytes and platelets [79]. Platelet α-granule constituents adhere and deposit high concentrations of pro- and anti-angiogenic regulatory factors, including PGFs, angiopoietins (Angs), and platelet microparticle proteins (PMPs) in a localized manner to promote cell–cell interactions and ultimately stimulate vascular repair [80]. Platelet angiogenic inhibitors, including endostatin, platelet factor-4, thrombospondin-1, A2-macroglobulin, plasminogen activator inhibitor-1, and angiostatin [81], inhibit angiogenesis by a differential release during angiogenic cascades [15,76].

It has been assumed that the contents of α-granules are homogeneous; however, the mechanisms of how the selective release of these granules regulates angiogenesis (stimulates or inhibits) are confusing. Interestingly, Italiano et al. reported that pro-angiogenic and anti-angiogenic proteins are segregated into different sets of platelet α-granules [15]. Their study suggests that at least two populations of α-granules containing endogenous angiogenic regulatory proteins are present in platelets and raises the possibility that platelets may contain multiple types of α-granules. The phenomenon that platelets do not release all of their granular contents when incorporated into a clot was also demonstrated by a morphometric evaluation of platelet releases during clot formation [82].

### 5.4. Pro-Angiogenic Platelet Factors

Numerous PRP platelet angiogenesis factors, including concentrated PGFs, are involved in angiogenesis [83]; however, their molecular mechanisms in blood vessel formation have not yet been completely elucidated [14]. Figure 4 presents an overview of pro- and anti-angiogenic platelet constituents.

#### 5.4.1. VEGF

VEGF is the most potent angiogenic stimulator and is one of the most well-studied PRP platelet growth factors implicated in the migration, mitogenesis, sprouting, and tube formation of ECs. The VEGF family is composed of five principal ligands (VEGF-A, -B, -C, and -D, and PLGF, a placenta-derived growth factor), the viral genome-derived VEGF-E, and three receptors (VEGF-R1, -R2, and -R3) [84]. VEGF-A and -B greatly impact blood vessel growth via VEGFR-1 and VEGFR-2 [84], whereas VEGF-C and -D stimulate lymphatic angiogenesis via VEGFR-3 and regulate early embryonic angiogenesis to a lesser extent. VEGF-A is multifaceted, including its ability to enhance angiogenesis, vascular permeability, and cell migration in macrophages and ECs. Most importantly, VEGF-A attracts ECs and promotes their differentiation, proliferation, and survival. In PRP preparations, the VEGF concentration is significantly increased compared to its whole blood values but fluctuates largely (30–558 pg/mL), depending on the devices used, the activating agent, the number of spin cycles, the duration of the spins, and, most importantly, the whole blood preparation volume [85,86,87].

#### 5.4.2. Platelet-Derived Growth Factor-BB (PDGF-BB)

PDGF is an important regulator of angiogenesis and exists as functional homo- and heterodimers of isotypes PDGF-A, PDGF-B, PDGF-C, PDGF-D, and heterodimer PDGF-AB [88]. All members of the PDGF family have potent angiogenic activity in vivo. A crucial role for PDGF-B and PDGF receptor-β has been demonstrated in angiogenesis by increased vascularity and maturation of the vascular wall. Moreover, PDGF stimulates EC proliferation and VEGF secretion [89].

PDGF is recognized as one of the most important factors regulating angiogenesis. Much PDGF data have been generated in myocardial revascularization research. For example, PDGF-AB promotes angiogenesis and reduces infarct size [90]. In a comprehensive review of MSK pathologies, Chen et al. summarized the role of PDGF-BB in tendon pathologies; PDGF-BB has a positive effect on tendon healing by enhancing inflammatory responses, promoting angiogenesis, and increasing collagen synthesis and the biomechanics of the repaired tendon [91]. Interestingly, Kovacivic et al. showed a dose-dependent response in angiogenesis using different recombinant PDGF-BB doses [92].

#### 5.4.3. Transforming Growth Factor-Beta (TGF-β)

TGF represents two classes of polypeptide platelet growth factors: TGF-α and TGF-β. These two classes are not structurally or genetically related to one another, and both act through different cell membrane receptor mechanisms. The most known class is TGF-β, with crucial roles in cell differentiation, immune regulation, and regenerative tissue repair. TGF-β is abundant in platelet α-granules as subtypes TGF-β1, -2, and -3, which belong to the TGF superfamily comprised of over 30 similar growth factors [16]. Cell signaling is regulated by three major receptors, also known as activin receptor-like kinases (ALK)—TGF-β receptor type I (TGF-βRI), type II (TGF-βRII), and type III (TGF-βRIII) [93].

In vivo, TGF-β has multiple functions and pathways promoting ECM deposition and integrin receptor upregulation. Furthermore, TGF-β regulates angiogenesis by modulating EC proliferation, migration, and tube formation [8] by alternating several ALK cascades. Additionally, TGB-β promotes its own expression, upregulates PDGF, bFGF, and TNF-α, and reverses VEGF from pro-survival to pro-apoptosis [94,95].

#### 5.4.4. Basic Fibroblast Growth Factor (bFGF)

bFGF, also known as FGF-2, is a granule-derived growth factor first identified as a pro-angiogenic molecule (5-cross). Currently, the FGF family comprises twenty factors and four receptors (FGFR-1,-2,-3, and -4), collectively mediating a broad range of biological effects [96].

In contrast to most growth factors, FGF-1 and FGF-2 have their own extracellular support systems that play a role in angiogenesis; most growth factors are secreted by the cells in which they are generated. An important characteristic of FGF is its ability to bind to heparan sulfate proteoglycans found on the surface of most cells and within the extracellular matrix [97]. In angiogenic pathways, FGF-2 stimulates the secretion of proteases and plasminogen activators by ECs, ultimately leading to the degradation of blood vessel basement membranes [98]. The ECs then migrate into the surrounding matrix, where they differentiate, proliferate, and form new blood vessels [99]. Following the formation of a new basement membrane, ECs secrete PDGF, attracting pericytes, which subsequently ensures the stability of the newly formed vessels [71].

#### 5.4.5. Angiopoietins (Ang)

Angiopoietins 1–4 (Ang1–4) are a family of growth factors that signal through tyrosine kinase receptors, Tie1 and Tie2; they are important for maintaining vessel integrity and endothelial permeability [100]. Ang1–4 bind to EC-specific Tie receptors [16]. The best-characterized angiopoietins are Ang1 and Ang2. Ang1 is stored in pericytes, smooth muscle cells, and in large quantities in platelets [101]. Nachman et al. identified the importance of platelet Angs as pro-angiogenic factors [102]. The number of circulating platelets and the platelet dose in PRPs influence the level of Ang1 in angiogenesis. Ang1 acts as a potent angiogenic growth factor signaling through Tie2 and plays a key role in angiogenesis in vivo, with distinct VEGF functions [103]. The function of Ang2 is to antagonize Ang1 and prime the effects of inflammatory cytokines and VEGF [104].

#### 5.4.6. Platelet Microparticles (PMPs)

PMPs are small fragments exfoliating platelet granules after platelet activation or apoptosis. Typically, PMPs circulate in peripheral blood and contain a unique subset of platelet proteins and lipids [105]. PMPs are involved in several biological processes, such as hemostasis, inflammation, and angiogenesis. The PMP surface lipid component stimulates the proliferation, migration, and tube formation of HUVECs [106]. Furthermore, Brill et al. showed that, depending on the presence of platelet growth factors, VEGF, bFGF, and PDGF in vivo, PMPs induce angiogenesis by provoking sprouting and the invasion of ECs [83]. Like the PGF, PMPs stimulate the growth of endothelial progenitor cells via intracellular kinase receptor signaling. Angiogenesis is further stimulated by interactions between PMPs and endothelial progenitor cells mediated by the expression of platelet membrane receptors CD62P (P-selectin) and glycoproteins IIb/IIIa and Ib [107].

#### 5.4.7. Serotonin (5-HT)

Serotonin (5-hydroxytryptamine, 5-HT) is a well-known neurotransmitter that plays critical roles in the central nervous system, particularly in pain tolerance, mood, and sleep [108]. 5-HT also plays important roles in peripheral tissues. It regulates primary homeostasis, gut motility, immune responses, vasoconstriction, and has stimulatory or inhibitory immune-modulating capacities through specific 5-HT receptors [109].

In humans, the majority of peripheral 5-HT is stored in the platelet dense granules, among other constituents [3]. During platelet activation and inflammation, colossal amounts of 5-HT are released from PRP [110]. 

Surprisingly, 5-HT is less-known for its role in angiogenesis. Released 5-HT has a stimulatory effect on the proliferation, migration, and tubulogenesis of endothelial cells [111,112]. The pro-angiogenetic effect of 5-HT in human endothelial cells was validated in a study by Zamani and Qu [113]. They concluded that 5-HT is a very potent angiogenic factor, driving angiogenic pathways via phosphorylation signaling in endothelial cells.

### 5.5. Pro-Angiogenic Leukocytes

As a result of homeostasis in physiological and chronic pathological conditions, neutrophils, monocytes, and tumor necrosis factor-alpha (TNF-α) are recruited to tissues from the peripheral blood to interact with macrophages and microvascular cells, promoting angiogenesis [114,115].

#### 5.5.1. Neutrophils

Neutrophils, monocytes, and dendritic cells are innate immune cells in peripheral blood. Neutrophils are essential leukocytes that are active in numerous healing pathways that create dense barriers against invading pathogens in conjunction with anti-microbial proteins that are present in platelets [116,117]. In orthobiology and regenerative medicine applications using neutrophil-rich PRP preparations (NR-PRP), platelet–leukocyte interactions regulate inflammation, wound healing, and tissue repair via the Toll-like receptors (TLRs). Specifically, TLR-4 on platelets stimulates interactions between neutrophils and platelets, resulting in the formation of neutrophil extracellular traps (NETs) [118] to trap bacteria and kill them by NETosis [52]. Importantly, Aldabbous et al. implicated NETs in angiogenesis, providing a link between NETs and inflammatory angiogenesis in vitro and in vivo [119]. This group demonstrated the role of NETs in regulating pro-inflammatory and pro-angiogenic responses in human ECs mediated by TLR-4 signaling. A pro-angiogenic response of neutrophils was also confirmed by Fetz and co-workers [REF]. Neutrophil MMP-9 is an endogenous matrix component that activates VEGF, initiating angiogenesis in biomaterial vascularization and in in situ tissue regeneration [120]. Furthermore, in a corneal injury model, immunohistochemistry suggested that corneal epithelial cells and infiltrating neutrophils express VEGF; however, no VEGF signaling was detected in the ECs after neutrophil depletion, signifying that neutrophils are not only capable of producing VEGF but can also interact with ECs to stimulate VEGF production by these cells [121].

#### 5.5.2. Monocytes, Macrophages, and TNF-α

Despite their critical role in the immune system and tissue repair, monocytes are less well known for how they participate in angiogenesis, considering a high percentage of endothelial progenitor cells (EPCs) circulating in the blood are of monocytic origin [122]. A study conducted by Shansila et al. demonstrated that early onset EPCs consist primarily of monocytes and T cells, indicating that the formation of EPCs is highly dependent upon the presence of monocytes [123]. Monocytes constitute most circulating cells expressing VEGF-R [124].

It is well established that macrophages have a range of phenotypes, from M1 macrophages that are classically activated to M2 macrophages that are alternatively activated, including their subsets M2a and M2c. As part of the healing cascade, these macrophage phenotypes play an important role in regulating vascularization through the secretion of specific cytokines [125]. M2 macrophages are normally regarded as pro-angiogenic and tissue repair cells.

In addition to secreting potent angiogenic stimulants, Spiller et al. found that M1 macrophages secrete the chemoattractant PDGF-BB, which may stabilize pericytes and facilitate the sprouting of ECs. Moreover, M2 macrophages secrete high levels of MMP-9 and are involved in vascular remodeling [126]. Activated M1 macrophages synthesize various inflammatory cytokines, including TNF-α, which activate local ECs and support leukocytic cell recruitment [127]. A switch is then made to tissue repair processes, simultaneously initiating the angiogenic healing cascade [128]. Sainson et al. linked TNF-α to priming ECs for angiogenic sprouting and stimulating ECs to recruit pericytes [129].

### 5.6. Pro-Angiogenic Plasma-Based Growth Factors

In addition to PRP platelet angiogenic effectors, the plasma component of PRP contains additional growth factors, including insulin-like growth-1 (IGF-1) and hepatocyte growth factor (HGF), which are part of the angiogenic development pathways.

#### 5.6.1. Insulin-Like Growth Factor (IGF-1)

The vascular IGF-1 system includes IGFs, their IGF-1 receptors (IGF-1R), and numerous proteins that carry out multiple physiological effects on the vasculature via endocrine, autocrine, and paracrine mechanisms [130]. Nakao-Hayashi et al. demonstrated the stimulatory effects of IGF-1 on vascular EC migration, tube formation, and the prevention of apoptosis [131].

Growing evidence indicates a role for IGF-1 in angiogenesis, cell proliferation, and diabetic vascular disease [132]. Both IGF-1 and IGF-1R are crucially important for retinal angiogenesis [133]. For ischemic and poorly vascularized tissues in MSK pathologies, monocyte expression of IGF-1 is involved in inflammation-linked angiogenic processes [134].

#### 5.6.2. HGF

HGF is primarily synthesized by hepatocytes as a mesenchyme-derived mitogen and circulates abundantly during plasma-stimulated cell migration. It is also involved in the branching tubular morphogenesis of epithelial cells and ECs and induces angiogenesis [135]. HGF exhibits strong mitogenic properties that regulate the development and regeneration of tissues [136]. In vivo experiments by Xin et al., they demonstrated that the combination of HGF and VEGF increases neovascularization in the rat corneal assay greater than either growth factor alone, suggesting that the two factors act synergistically and provide a more effective strategy for stimulating tubulogenesis through paracrine HGF secretion and amplifying the angiogenic response [1]. More recently, Zhang and co-workers demonstrated that HGF stimulates the secretion of soluble secretory products by tendon stem cells that promote the repair and functional recovery of ruptured Achilles tendons, revealing a strong connection between HGF and VEGF-A, where HGF might stimulate the release of VEGF-A [137].

### 5.7. In Vitro and Animal Studies

Many in vivo and in vitro studies have shown that different PRP constituents (e.g., platelet growth factors, cytokines, Angs, and various leukocytes) play pivotal effector roles in regulating angiogenesis, matrix formation, and cell proliferation during tissue repair [91]. Bertrand-Duchesne et al. studied the mechanisms of various PDGF fractions in PRP that stimulate angiogenesis [138]. Their analysis revealed high levels of VEGF, PDGF-BB, epidermal growth factor, and basic fibroblast growth factor (bFGF) in PRP. They used human umbilical vein ECs (HUVECs) for proliferation assays to evaluate the effect of these growth factors in the PRP supernatants. Another study found that the PRP formulations increase human microvascular endothelial proliferation, migration, and differentiation [59]. Additionally, Li et al. observed that PRP promotes vascular growth and stimulates endothelial progenitor cells to form vessel-like structures [139]. Moreover, a fibrin-based PRP matrix increased cell proliferation while decreasing apoptosis in HUVEC and skeletal myoblast cultures [140]. A recent study confirmed that concentrated growth factors containing pro-angiogenic factors (VEGF, TGF-1, MMP-2, and MMP-9) induce EC migration, tubule formation, and endothelial expression of multiple angiogenic mediators. Furthermore, PRP concentrations express endothelial CD34 stem cell markers, resulting in neoangiogenesis [141].

Zhou and co-workers investigated the effect of PRP-gel on open abdominal wounds in rats, demonstrating higher blood perfusion in the original lesion and more mature granulated tissue when compared to those treated with platelet-poor plasma [142]. In addition, an injection of PRP-gel within the injured muscle tissue of mice induced the reperfusion of blood into the lesion. Moreover, the release of PDGFs in a gelatin hydrogel containing PRP to stimulate wound healing showed increased epithelialization and confirmed vascular growth compared to other treatments [143]. Roy et al. showed that a platelet-rich fibrin matrix stimulated wound healing in a porcine model by enhancing angiogenesis [144]. Furthermore, when Achilles tendons were sectioned in New Zealand rabbits to stimulate tendon regeneration and subsequently treated with PRP or saline, the PRP-treated tendons showed increased levels of angiogenesis and better collagen fiber re-alignment compared to saline-treated specimens [145]. To summarize, various studies have shown that PGF fractions in PRP may exert a potent pro-angiogenic response in different anatomical locations.

### 5.8. PRP Classification

There is no generally accepted standard or classification system in place for PRP devices and technologies [146]. Rossi et al. mentioned three principal groups of PRP preparations for orthobiological applications: leukocyte-rich (LR)-PRP, leukocyte-poor (LP)-PRP, and pure platelet-rich fibrin (P-PRF).

Although LR-PRP and LP-PRP are more specific than the generic PRP definition, they do not contain any specific details regarding the presence of specific types of leukocytes or their concentrations [147]. Notably, PRP and platelet lysates (PLs) differ in their PGF content, cytokine and chemokine composition, and contamination with leukocytes due to different preparation protocols [148].

Furthermore, PRP and PRP-G may be administered in a variety of ways, and the optimal method depends on the pathoanatomic tissue pathology and the treatment objectives [149]. Ultrasounds and fluoroscopic guidance should be used in order to ensure the precise delivery of the biological products.

#### 5.8.1. Leukocyte-Poor PRP Angiogenic Potential

In most instances, LP-PRP is classified as a platelet concentrate without a significant influx of leukocytes in the final PRP product, particularly neutrophils [150]. It has been suggested that a non-significant inflammatory tissue response is to be expected following LP-PRP administration [151]. Consequently, the release of cytokines and growth factors that induce fibroblast chemotaxis and angiogenesis is minimal, with nominal collagen synthesis [152], negatively impacting ECM–cell interactions by regulating angiogenesis and vascular remodeling [47]. This hypothesis was confirmed by Yuan and co-workers, who found that LP-PRP induced significantly fewer angiogenic responses but was more effective in regulating angiogenesis [153].

#### 5.8.2. Leukocyte-Rich PRP Angiogenic Potential

Autologous LR-PRP concentrates are a rich source of heterogenous cells, including concentrated platelets, a buffy-coat layer containing neutrophils, monocytes, lymphocytes, and leukocytes, and other biomolecules that are directly or indirectly involved in angiogenesis and tissue repair [154]. Activated leukocytes release various types of proteases, including MMPs [155]. More specifically, MMP-2 and MMP-9 participate directly in angiogenic pathways, as they play a significant role in the survival, proliferation, and migration of endothelial cells. Hence, platelets also release MMPs on a platelet-dose-dependent basis; the total concentration of MMPs increases and, therefore, their contribution to angiogenesis is increased [156]. Furthermore, the presence of concentrated leukocytes in PRP affects the concentration of cell cytokines and PGF levels [157]. Specifically, monocytes can secrete significant pro-angiogenic molecules, including the expression of VEGF [158].

Kobayashi et al. showed that a combination of peripheral blood leukocytes and platelets induced angiogenesis [155], which was recently confirmed by other studies [3]. Leukocytes in PRP promote cell chemotaxis, proliferation, and differentiation, chemotaxis recruits neutrophils, macrophages, and lymphocytes to tissue injury sites. These inflammatory cells release another wave of pro-inflammatory cytokines (e.g., TNF-α and interleukins) and growth factors (FGF and VEGF) that stimulate angiogenesis, promoting tissue regeneration and driving the transition to the proliferative phase of healing [159]. Yuan et al. found that angiogenesis is more pronounced with LR-PRP than LP-PRP, with higher levels of VEGF and α-SMA expressed during tissue repair [153]. Additionally, increased TGF-β1 release promoted α-SMA protein expression, reflecting the number of functional arterioles and myofibroblasts [160]. Most importantly, their study indicated that the blood supply was more abundant in the LR-PRP group, which is crucial for tissue repair.

#### 5.8.3. PRF

Key determinants in tissue repair and regeneration are the biological properties of the applied orthobiological platelet suspensions to support in angiogenesis and tissue repair. However, conflicting clinical outcome data have been reported (Table 2), most likely due to inconsistencies in platelet concentrate preparation protocols and the possibilities of implementing platelet dosing and formulation strategies [161]. Table 3 indicates critical differences between high-volume PRP devices and low volume PRF tubes, which prepare a platelet suspension from the acellular plasma layer, excluding erythrocytes and in most products, excluding leukocytes as well [66].

It is fair to assume that low volume and single-spin PRF tubes produce platelet treatment specimens with low tissue repair and angiogenetic capacities when compared to high-volume double-spin PRP devices, as demonstrated by Oh et al. who measured overall higher platelet growth factor concentrations following double-spin preparations [162].

**Table 3 biomedicines-11-01922-t003:** An overview of some significant and non-significant treatment outcomes following PRP applications in soft tissue MSK disorders, with emphasis on platelet dose and bioformulation. Abbreviations: PRP: platelet-rich plasma; LP: leukocyte poor; LR: leukocyte rich; ACL: anterior cruciate ligament.

Study Identifier	PRP Application	PRP Dose × 10^9^ Platelets	Bioformulation LP/LR	Reference
	Non-significant outcomes			
1	Rotator cuff repair	0.55	LP	[163]
2	Lateral epicondylitis	0.60	LP	[164]
3	Rotator cuff repair	1518	LP	[165]
4	Patella tendinopathy	0.663	LP	[163]
5	Hamstring tendinopathy	0.750	LP	[166]
6	Rotator cuff repair	0.701	LP	[167]
7	Achilles tendinopathy	0.875	LP	[168]
8	Shoulder soft tissue	1312	LR	[169]
9	Achilles tendinopathy	1313	LP	[170]
10	Achilles tendinopathy	1462	LR	[171]
11	Rotator cuff repair	1575	LP	[172]
12	Achilles tendinopathy	2430	LR	[173]
13	Lateral epicondylitis	3037	LR	[174]
14	Achilles tendinopathy	3125	LR	[175]
	Significant outcomes			
15	Lateral epicondylitis	2454	LR	[176]
16	Rotator cuff repair	3000	LR	[177]
17	Lateral epicondylitis	3877	LR	[178]
18	Ulnar collateral ligament	3900	LR	[179]
19	Lateral epicondylitis	4500	LR	[180]
20	Lateral epicondylitis	4500	LR	[181]
21	Rotator cuff surgical repair	8100	LR	[38]
22	Patellar tendinopathy	5100	LR	[182]
23	ACL repair	4500	LR	[183]
24	Plantar fasciitis	3500	LP	[184]
25	Rotator cuff repair	5.275	LR	[185]

#### 5.8.4. PLs

PLs are manipulated, non-freshly prepared, and concentrated platelet injectates. In PLs, GFs are intentionally extracted from the platelets using various additional preparation techniques [186]. The objective when preparing PLs is to disrupt the cellular membranes of platelets, releasing all PGF and angiogenic factors. The rationale for using PLs is that treated tissues respond faster to high levels of PGF that are instantly available compared to non-activated PRP platelets, which must degranulate before PGF is released [187]. Nevertheless, PGF in PLs has a shorter half-life, negatively affecting EC proliferation and impairing adequate endothelial tube formation, as demonstrated in an in vitro study using HUVECs [188].

## 6. The Angiogenic Potential of PRP Is Contingent on Platelet Dose and Bioformulation

PRP fractions are frequently used in tissue repair and regenerative procedures, as activated platelets release over 300 biologically active factors [76]. Depending on the application techniques and the presence of anticoagulants, freshly prepared autologous PRP can be used in liquid or gelatinous forms. In leukocyte-enriched PRP preparations, leukocytes may cause significant cellular and tissue effects because their immune and host defense mechanisms can influence the intrinsic biology of acute and chronic tissue conditions [118,189,190]. Activated PRP releases a profuse amount of platelet constituents, triggering various physiological reparative cascades, including angiogenesis by differential exocytosis of angiogenic platelet activators and inhibitors [191] (Figure 5). Unfortunately, little is known about the angiogenic differences and capacities of different PRP bioformulations.

In the field of orthobiological non-surgical PRP procedures for treating chronic and degenerative pathologies, angiogenic microvascular networks are essential to biological repair processes. Successful restoration of microvasculature networks reconstitutes the exchange of nutrients, oxygen, and waste products, facilitating multiple local cellular functions required for tissue repair [11,192].

The optimal platelet dose and bioformulation, specific to each tissue type and pathology, are key factors in maximizing the results of PRP therapies. Regrettably, different PRP systems utilize different preparation protocols, resulting in PRP specimens consisting of different quantities of platelets, with or without particular leukocytes [162]. In general, neutrophils are inflammatory cells, triggering post-PRP treatment flares, whereas monocytes and macrophage phenotypes are reparative cells that support various actions in the concentrated platelets [3]. What is less apparent is that multicellular PRP products containing a high dose of platelets, neutrophils, monocytes, and macrophages are actively involved in angiogenic processes in chronic, pathological tissue [114,193].

Many degranulated platelet constituents and leukocytes contribute to (neo)angiogenesis to re-establish the microvasculature in pathological, degenerated tissues. (A) Diminished angiogenesis because of chronic inflammation or a dysregulated local immune system during the initial repair process, with overexpression of the inflammatory mediators, such as ILs, TNF-α, and IFN-γ, the presence of degranulating mast cells, T cells, and Th cells, and the production of MMP-2 by macrophages. (B) After the administration of non-activated LR-PRP, an extensive tissue repair process is initiated. Activated platelets change their shape and develop pseudopods. Subsequently, they release their granular contents, and platelet mediators orchestrate a magnitude of biological events, including the release of pro-angiogenic mediators, PGF, PMPs, and 5-HT. VEGF-A is the most potent PGF, playing a pivotal role in angiogenesis, whereas PDGF-BB stabilizes pericytes and facilitates the sprouting of ECs. Released Angs are important in maintaining blood vessel integrity and endothelial permeability. Furthermore, the released platelet constituents recruit cells and contribute to the proliferation and differentiation of the residing tissue cells. Platelets induce a platelet–neutrophil reaction, where neutrophils are responsible for NETosis to form NETs, linking inflammation to angiogenesis. Furthermore, specific monocytes (M1 and M2) and neutrophils influence the activities of ECs to form new microvasculature networks in existing blood vessel structures. Additionally, IGF-1 and HGF plasma-based growth factors stimulate VEGF-A release and tubulogenesis via EC migration and prevent apoptosis. © Following adequate platelet and leukocyte actions, sufficient angiogenesis is accomplished in pathological tissues, supplying them with oxygen and nutrients and removing waste products during tissue repair. Furthermore, a revised vascular network enables the delivery of MSCs to the area to support the reparative process. Abbreviations: MC: mast cell; IL: interleukin; Th: T helper cell; NEU: neutrophil; MO: monocyte; M1: macrophage phenotype 1; M2: macrophage phenotype 2; NET: neutrophil extracellular trap; PC: pericyte; EC: endothelial cell; IGF-1: insulin-like growth factor 1; HFG: hepatocyte growth factor; TGF-β1: transforming growth factor beta 1; VEGF-A: vascular endothelial growth factor-ligand A; SDF-1α: stromal cell-derived factor-1 alpha; Ang-1: angiopoietin-1; IL-8: interleukin-8; MMP: matrix metalloproteases; PMP: platelet microparticles; TNF-α: tumor necrosis factor-alpha; INF-γ: interferon-gamma; 5-HT: serotonin; PDGF-BB: platelet-derived growth factor-BB; bFGF: basic fibroblast growth factor.

The fact that angiogenesis is an integral part of tissue repair and regeneration and that large variations in platelet numbers and bioformulations have been observed among currently available PRP products makes it tempting to speculate whether most of these devices can restore the microvasculature since PRP stimulates cell–cell interactions in a dose-dependent manner. A low PRP platelet dose will probably have a minimal impact on angiogenic processes and tissue repair [194]. Moreover, Magalon and associates described a positive correlation between platelet dose and the quantity of PGF and other platelet constituents [195]. Figure 6 is a graphic impression to visualize patient outcomes of several orthobiologic studies in which a variety of PRP dosing regimens were used, illustrating significant positive patient outcomes as well as non-significant outcomes. Table 3 portrays more in detail the treated MSK disorders in which PRP was applied, with mention of the platelet dose and bioformulation of each study.

Based on this narrative review, it is most likely that the PRP angiogenic potential in the non-significant outcome studies was insufficient to induce angiogenesis to stimulate tissue repair processes.

## 7. Discussion

During non-surgical orthobiological and other regenerative medicine therapies, clinical PRP preparations allow the delivery of an abundance of different cells and biomolecules released by activated platelets at pathoanatomic tissue sites. This delivery prompts various physiological cascades, promoting on-site immunomodulation, inflammatory processes, and angiogenesis to initiate healing and tissue repair [196]. The rationale for employing PRP in MSK pathologies includes the precise and local delivery of a sufficient PRP platelet dose containing non-activated platelets with granules encompassing many PGF, cytokines, chemokines, and other molecules. PRP specimens can also incorporate leukocytic cell concentrations, depending on their bioformulations and preparation principles.

PRP procedures demonstrate mostly promising results following orthobiological treatments. Unfortunately, the literature also reports studies lacking positive patient outcomes or having negative effects on outcomes. It has been postulated that the inability to restore an adequate microvascular blood vessel network in chronic pathological and inflammatory tissues is the main reason for negative treatment outcomes [197,198]. Remarkably, re-establishing angiogenesis is even more important in areas of poor vascularization, such as meniscal tears and tendon and ligament injuries [199,200]. Interestingly, in these poor outcome studies, the injected PRP products had a low number of total platelets, and, in most instances, leukocytes were not part of the PRP formulation. 

Researchers and physicians agree that tissue repair in chronic inflammatory pathologies and tissues with inadequate blood supply can only be accomplished once the local microvasculature network has been restored. When the milieu of pathological and degenerative microenvironments (low oxygen tension, low pH, and high lactate levels) cannot be rehabilitated, tissue repair and functional tissue healing will ultimately be diminished [201,202]. Therefore, the angiogenesis cascade must be initiated early and completed to successfully stimulate angiogenesis following PRP therapy for pathoanatomical tissue repair. In this regard, Kobayashi et al. demonstrated in an experimental model that the local application of a high concentration of LR-PRP improved tissue healing through the recruitment of reparative cells via the blood flow and through tissue repair cascades in the local PRP-administered microenvironment [155]. These observations were confirmed in an independent study, indicating that tissue repair is only facilitated when adequate angiogenesis is ensured, because this process is crucial for delivering oxygen and nutrients and removing catabolic byproducts to match the high metabolic activity during tissue repair [203]. From this perspective, Andia et al. mentioned that the number of tenocytes increase in parallel with an increase in angiogenesis in tendon injuries [204].

Platelets play key roles during the initiation of angiogenesis and its amplification phases. The serum levels of platelet growth factors increase following platelet activation. Furthermore, platelets interact with macrophage phenotypes and other leukocytes, releasing angiogenic factors and stimulators to facilitate the sprouting and organization of micro-vessels from preexisting blood vessels [12]. As such, we strongly believe that high platelet doses in clinical LR-PRP preparations are essential to implementing orthobiological treatment strategies, because platelet and leukocytic constituents interact intensely with angiogenic activities, stimulating (neo) angiogenesis to re-establish the microvascular architecture in pathological tissue structures. Indeed, Dittadi and colleagues demonstrated a highly significant correlation between serum VEGF concentrations and platelet doses in a controlled study [205]. Furthermore, several studies have investigated the effect of PRP applications during ligament surgery, using MRI imaging [206,207]. A significant increase in the levels of vascularity, building up to higher rates of ligamentization, was observed by both groups at 6 months post therapy.

The optimal platelet concentration for promoting angiogenesis was determined in a platelet dose-defining study by Giusti and co-workers, who studied the effects of activated platelet concentrations on the angiogenic potential of HUVECs [77]. They concluded that the optimal platelet concentration was 1.5 × 10^6^ platelets/µL, whereas lower or higher platelet concentrations had a significantly lower angiogenic potential. As an explanation, injecting 5 mL of PRP with the optimal concentration delivers 7.5 billion platelets to a particular tissue site. A flawless dose–response relationship between the platelet count and the extent of sprouting angiogenesis was validated [208].

In addition to platelets, leukocytes may also be included in PRP formulations, depending on the PRP device used. A prominent function of leukocytes in angiogenesis is their recruitment to areas of injury and hypoxia in large numbers. At pathological tissue sites, the recruitment of monocytes involves various chemokines acquiring pro-angiogenic properties, and macrophages secrete collagenase, TGF, and PDGF [115,209,210]. Furthermore, macrophages can release the angiogenic stimulators IL-1, FGF, and TNF-α [193]. Moreover, when M1 macrophages, leukotriene B4, and platelet-activating factor interact with ECs, neutrophils and lymphocytes adhere to the ECs and migrate away from them [211].

An orthobiological PRP formulation incorporating neutrophils can be important in initiating angiogenic processes by releasing VEGF [211] and the pro-angiogenic enzyme MMP-9 [212]. Subsequently, MMP-9 degrades the ECM, releasing vast amounts of matrix-bound VEGF-A [213,214].

Given all the possible variables in PRP compositions and devices, LR-PRP should be prepared from a unit of whole blood using devices that capture a high number of platelets to minimize the total amount of a patient’s blood required. In this manner, LR-PRP can be prepared with a large number of platelets from which several high platelet dosages can be generated to treat multiple tissue sites while effectively preserving the regulation of a significant number of platelet angiogenic factors that facilitate blood vessel formation and tissue repair [25,215]. 

Further studies are warranted to understand optimal platelet dosing strategies that address tissue specificity (e.g., tendons, ligaments, and cartilage), the chronicity of the disorder, and the severity of pathological conditions (like tendinopathy vs. tears). Finally, emphasis should be given to correlating platelet dosing and bioformulation strategies to patient outcomes in an attempt to standardize orthobiological PRP treatments.

## 8. Conclusions

In non-surgical, interventional repair of pathoanatomic tissues, a functioning microvascular network is a conditio sine qua non. However, under these chronic circumstances, the angiogenic process is profoundly reduced, hindering the supply of oxygen and nutrients required to initiate tissue repair. Orthobiological PRP bioformulations comprised of a high dose of platelets and leukocytes hold a plethora of angiogenic stimulators capable of inducing angiogenesis and restoring microvasculature networks, facilitating tissue repair and regeneration.

## Figures and Tables

**Figure 1 biomedicines-11-01922-f001:**
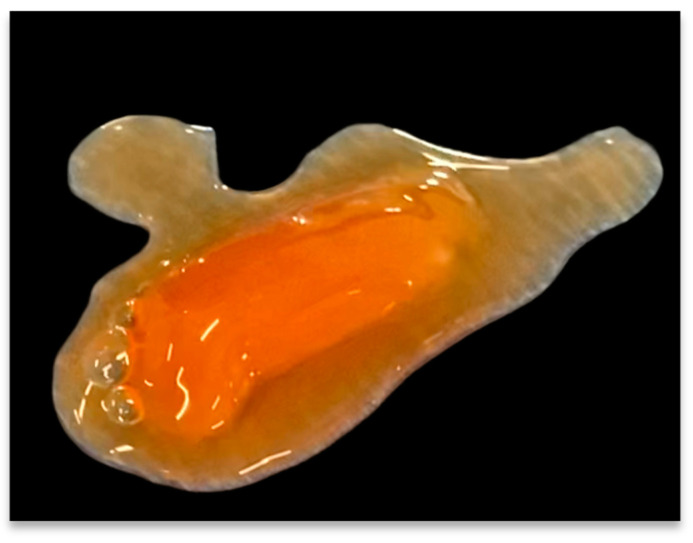
PRP-G clot. A liquid, 7 mL PRP concentrate specimen, was mixed with a combination of 0.30 mL of CaCl (10%) and 0.2 mL of thrombin (50 IU) to induce the polymerization of fibrinogen, which is present in the PRP plasma fraction, into fibrin. The result is a stable PRP-G scaffold that contains high concentrations of platelets and eventually leukocytic cells, when compared to PRF preparations [35].

**Figure 3 biomedicines-11-01922-f003:**
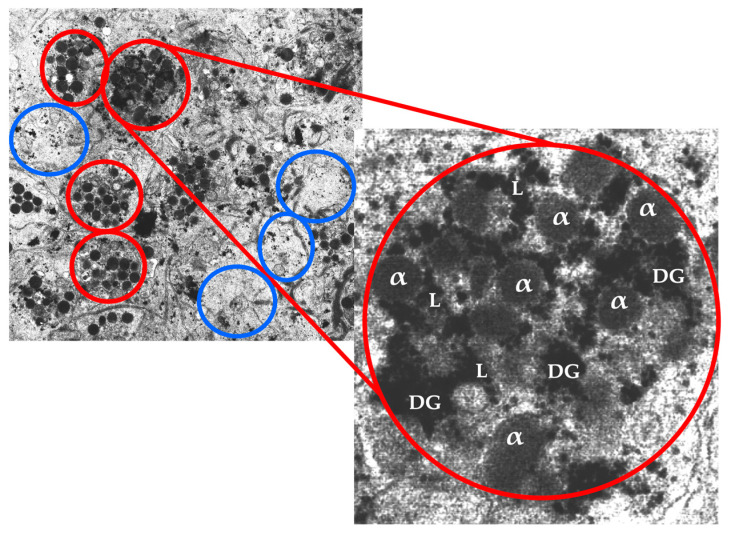
An electron microscopic image of an ultrathin (70 nm sample) PRP fraction. In the red circles, individual non-activated platelets are visible. In the blue circles, platelets have released their content following platelet activation. A single, non-activated, platelet at magnification ×7000 visualizes the three different platelet granules. Abbreviations: α: alpha granule; DG: dense granule; L: lysosome.

**Figure 4 biomedicines-11-01922-f004:**
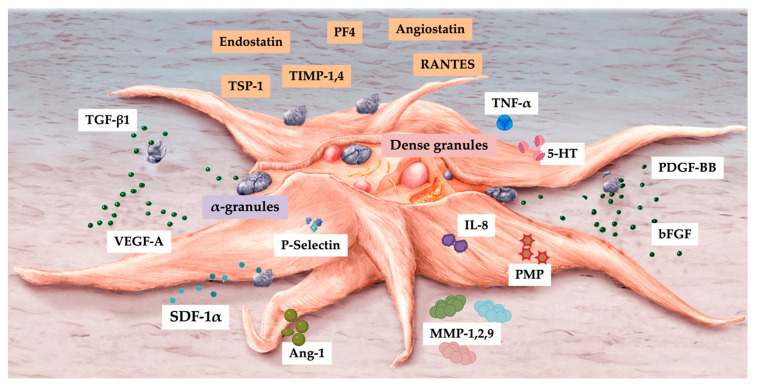
An illustration of an activated platelet, indicating pro- and anti-angiogenic platelet constituents released from α and dense granules. Abbreviations: TGF-β1: transforming growth factor beta 1; VEGF-A: vascular endothelial growth factor-ligand A; SDF-1α: stromal cell-derived factor-1 alpha; Ang-1: angiopoietin-1; IL-8: interleukin-8; MMP: matrix metalloproteases; PMP: platelet microparticles; TNF-α: tumor necrosis factor-alpha; 5-HT: serotonin; PDGF-BB: platelet-derived growth factor-BB; bFGF: basic fibroblast growth factor; TSP-1: thrombospondin 1; TIMP: tissue inhibitor of metalloproteinase; PF4: platelet factor 4; RANTES: regulated by T-cell activation and probably secreted by T cells.

**Figure 5 biomedicines-11-01922-f005:**
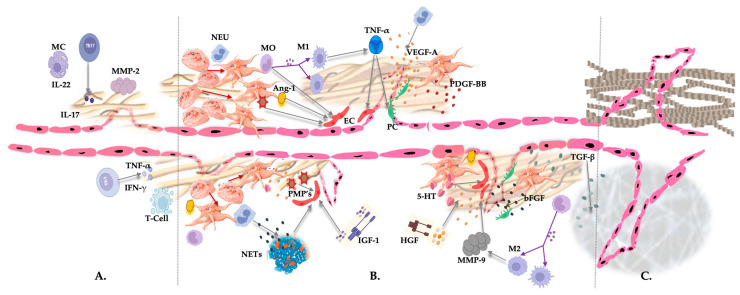
Platelet- and leukocyte-mediated angiogenic pathways.

**Figure 6 biomedicines-11-01922-f006:**
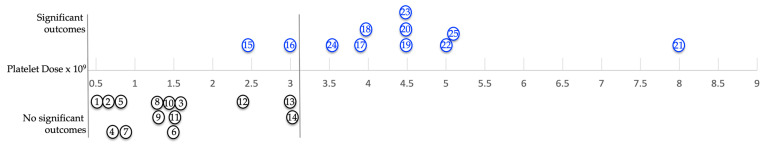
The effectiveness of PRP products was studied for a variety of orthobiological soft tissue pathoanatomic conditions, including tendinopathies, hamstring pathologies, meniscus lesions, and plantar fasciitis. Each of these studies, represented by the circles, has the platelet dose depicted on the *X*-axis (e.g., in study 24, the platelet dose was on average 3.6 × 10^9^ platelets for all treated patients). Based on the assessment scores between the treatment groups in each study, the *Y*-axis indicates whether the patient outcomes were positive or negative. It was found that results of studies with identifiers 1–14 were not significantly different when compared with control groups, particularly when platelet doses were less than 1.5 × 10^9^ platelets in the majority of these studies. All markers in blue demonstrated a positive outcome after PRP treatment. A notable characteristic of these studies is that the platelet dose was generally higher than that observed in studies without significant outcomes. Studies involving greater than 3.2 × 10^9^ platelets have generally reported more positive results. (Based on the data provided in each study, the platelet dose was calculated; however, if no platelet count was provided, the baseline count was assumed to be 250.000/µL for all studies. The platelet capture rates and concentration factors were determined using the methods described by Magalon et al. and Fadadu et al. [2,4]).

**Table 1 biomedicines-11-01922-t001:** Most cited pro-angiogenic and anti-angiogenic molecular regulators of angiogenesis. The presence of individual angiogenic molecules in LR-PRP is annotated with an *. Abbreviations: LR-PRP: leukocyte-rich platelet-rich plasma; 5-HT: serotonin; Ang: angiopoietins; bFGF: basic fibroblast growth factor; G-CSG: granulocyte-stimulating growth factor; HFG: hepatocyte growth factor; IGF-1:insulin-like growth factor; IL: interleukin; MMP: matrix metalloproteases; PDGF-BB: platelet-derived growth factor-BB; PMP: platelet microparticles; SDF-1α: stromal cell-derived factor-1; TGF-β1: transforming growth factor; TNF-α: tumor necrosis factor-alpha; VEGF: vascular endothelial growth factor; PA: plasminogen activator; PF-4: platelet factor 4; RANTES: regulated by T-cell activation and probably secreted by T cells TIMP: tissue inhibitor of metalloproteinase; TSP-1: thrombospondin 1.

Pro-Angiogenetic Stimulator	Presence in LR-PRP	Anti-Angiogenic Inhibitors	Presence in LR-PRP
5-HT	*	Angiostatin	*
Ang	*	Endostatin	*
bFGF	*	Heparinases	
G-CSF		IL-10,12	
HGF		Interferon α/β/γ	
IGF-1	*/-	PA	
IL-3, 8		PF-4	*
Macrophage		RANTES	*
MMP-2	*	TGF-β	*
MMP-9	*	Thrombospondin-1	*
Monocyte	*	TIMP1-4	*
Neutrophil	*	TSP-1	*
PDGF-BB	*	Vasculostatin	
PMP	*		
Progranulin			
TGF-β	*		
Thrombin			
TNF-α			
VEGF	*		

**Table 2 biomedicines-11-01922-t002:** Performance characteristics and strategic utilization differences between most PRP and PRF devices.

PRP	PRF
High blood volume preparation devices	Low blood volume preparation tubes
Double-spin technology	Single-spin technology
High platelet capture rates	Low platelet capture rates
High treatment volumes	Low treatment volumes
High total available platelets	Low total available platelets
Options for therapeutic platelet dosing strategies	No options for therapeutic platelet dosing strategies
High platelet growth factor concentrations	Low platelet growth factor concentrations
Leukocyte formulation options	No leukocyte formulations options

## Data Availability

The data are available upon request from the corresponding author.

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
