# Peer review of "Angiogenesis and Tissue Repair Depend on Platelet Dosing and Bioformulation Strategies Following Orthobiological Platelet-Rich Plasma Procedures: A Narrative Review"

_biomedicines, 2023, doi:10.3390/biomedicines11071922_

Round 1
Reviewer 1 Report
The authors described "Angiogenesis and tissue repair depend on platelet dosing and bioformulation strategies following orthobiological platelet-rich plasma procedures. A narrative review." As they mentioned, the angiogenic cascade and the commonalities with the classical healing cascade in tissue repair and their relationship to platelet dosing and bioformulation strategies in PRP preparations are infrequently discussed. Moreover, much remains unknown regarding method of administration, and long-term outcomes in all fields of surgery. So, this topic should be attractive for potential readers. I have some questions and suggestions to improve this manuscript.
1. As they described, standardization of PRP remains a problem. How about the use of PRP in surgery? I think PRP should have more powerful effects in combining surgery. Please add when and how to use PRP.
2. What is the difference between PRP and PRF (Platelet rich fibrin)? Fibrin sealants have been used for several decades as hemostatic agents to achieve wound healing. Please add the role and function of PRF.
Reviewer 2 Report
The manuscript of Everts et al. is an extensive review of the literature on PRP, focusing on the role of these therapeutic preparations in angiogenesis. The review is interesting because it provides the reader with a considerable amount of information both on a theoretical and practical level. The preparation techniques of PRP strongly impact its biological content and potential therapeutic action. Having adequate knowledge of the different techniques that can be used is very important for those who use these products. From this point of view, the manuscript is undoubtedly interesting. However, I believe that some parts of the work should be carefully reviewed and rewritten using more appropriate language. Although I am not a native speaker, it seems clear that different parts have been written by different authors, with the English language in some parts inadequate. Above all, the abstract and introduction should be reviewed, and the language should be carefully controlled.
Apart from these aspects, I believe the work can be published without requiring substantial revisions.
I have provided my comments on the English language discussing the manuscript. I strongly suggest revising the language to provide a more pertinent language.
Author Response
Dear Reviewer 2.
Thank you for your time to review our manuscript, your suggestions, and interests in the manuscript.We reviewed as per your suggestion the abstract and introduction again, but we could not see your suggestions highlighted in the manuscript we downloaded, unfortunately.
For your information, the manuscript was of course reviewed by our native authors, and we used an independent external proof-reading company. You find their details here below.
If appropriate, we would be happy to review your comments, if we were able to see them. Our apologies for no further action at this moment.
Best regards, and again, thank you for your expert review.
On behalf of the authors,
Peter Everts.
Please see the attachments

Round 2
Reviewer 1 Report
The authors revised the manuscript precisely.
I have one more question regarding surgical applications of PRP.
Yamakawa S et al. reported that standardization of PRP to expand its clinical use also remains a problem as the varying concentrations of platelets, growth factors, and leukocytes are possibly responsible for conflicting study results (Burns Trauma, 2019). How about the optimal method of administration of PRP, and long-term outcomes in all fields of surgery? Please add it.
Author Response
Dear Reviewer 1.
Thank you for your review of our updated manuscript.
We added the following paragraph at line 658:
Furthermore, PRP and PRP-G may be administered in a variety of ways, and the optimal method depends on the pathoanatomic tissue pathology and the treatment objectives [1]. Ultrasound and fluoroscopic guidance should be used in order to ensure the precise delivery of the biological products.
And included the reference you mentioned.
However, we feel that to comment and address adequately on your comment on long term outcomes, we would need significantly more time and literature review. We suggest to keep this as an alternative paper, beyond the scope of angiogenesis.
I hope these answers, respectfully, are sufficient regarding your questions.
Best regards,
Peter Everts